# Neural Relational Inference
# with Fast Modular Meta-learning

**Ferran Alet, Erica Weng, Tomás Lozano Pérez, Leslie Pack Kaelbling**
MIT Computer Science and Artificial Intelligence Laboratory
{alet,ericaw,tlp,lpk}@mit.edu

## Abstract

*Graph neural networks* (GNNs) are effective models for many dynamical systems consisting of entities and relations. Although most GNN applications assume a single type of entity and relation, many situations involve multiple types of interactions. *Relational inference* is the problem of inferring these interactions and learning the dynamics from observational data. We frame relational inference as a *modular meta-learning* problem, where neural modules are trained to be composed in different ways to solve many tasks. This meta-learning framework allows us to implicitly encode time invariance and infer relations in context of one another rather than independently, which increases inference capacity. Framing inference as the inner-loop optimization of meta-learning leads to a model-based approach that is more data-efficient and capable of estimating the state of entities that we do not observe directly, but whose existence can be inferred from their effect on observed entities. To address the large search space of graph neural network compositions, we meta-learn a *proposal function* that speeds up the inner-loop simulated annealing search within the modular meta-learning algorithm, providing two orders of magnitude increase in the size of problems that can be addressed.

## 1   Introduction

Many dynamical systems can be modeled in terms of entities interacting with each other, and can be best described by a set of nodes and relations. *Graph neural networks* (GNNs) (Gori et al., 2005; Battaglia et al., 2018) leverage the representational power of deep learning to model these relational structures. However, most applications of GNNs to such systems only consider a single type of object and interaction, which limits their applicability. In general there may be several types of interaction; for example, charged particles of the same sign repel each other and particles of opposite charge attract each other. Moreover, even when there is a single type of interaction, the graph of interactions may be sparse, with only some object pairs interacting. Similarly, relational inference can be a useful framework for a variety of applications such as modeling multi-agent systems (Sun et al., 2019; Wu et al., 2019a), discovering causal relationships (Bengio et al., 2019b) or inferring goals and beliefs of agents (Rabinowitz et al., 2018).

We would like to infer object types and their relations by observing the dynamical system. Kipf et al. (2018) named this problem *neural relational inference* and approached it using a variational inference framework. In contrast, we propose to approach this problem as a *modular meta-learning* problem: after seeing many instances of dynamical systems with the same underlying dynamics but different relational structures, we see a new instance for a short amount of time and have to predict how it will evolve. Observing the behavior of the new instance allows us to infer its relational structure, and therefore make better predictions of its future behavior.

Meta-learning, or learning to learn, aims at fast generalization. The premise is that, by training on a distribution of tasks, we can learn a *learning algorithm* that, when given a new task, will learn from

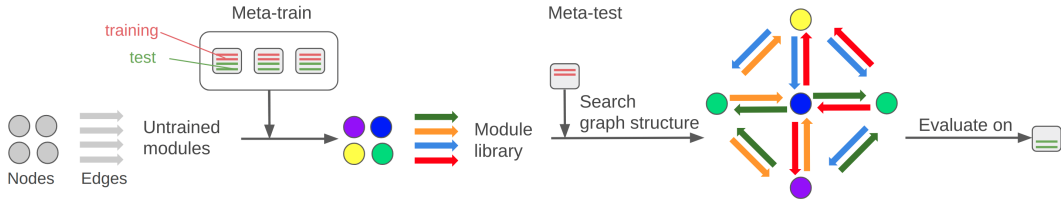

Figure 1: Modular meta-learning with graph networks; adapted from Alet et al. (2018). The system meta-learns a library of node and edge modules, represented as small neural networks; at performance (meta-test) time, it is only necessary to infer the combination of modules that best predict the observed data for the system, and use that GNN to predict further system evolution.

very little data. Recent progress in meta-learning has been very promising; however, meta-learning has rarely been applied to learn building blocks for a structured domain; more typically it is used to adapt parameters such as neural network weights. *Modular meta-learning* (Alet et al., 2018), instead, generalizes by learning a small set of neural network *modules* that can be composed in different ways to solve a new task, without changing their weights. This representation allows us to generalize to unseen data-sets by combining learned modules, exhibiting combinatorial generalization; i.e., "making infinite use of finite means" (von Humboldt, 1836/1999). In this work we show that modular meta-learning is a promising approach to the neural relational inference problem.

We proposed the `BounceGrad` algorithm (Alet et al., 2018), which alternates between simulated annealing steps, which improve the structure (the assignment of node and edge modules in the GNN) for each dataset given the current neural modules, and gradient descent steps, which optimize the module weights given the modular structure used in each dataset. This formulation of neural relational inference offers several advantages over the variational formulation of Kipf et al. (2018). Primarily, it allows joint inference of the GNN structure that best models the task data, rather than making independent predictions of the types of each edge. In addition, since it is model-based, it is much more data efficient and supports other inferences for which it was not trained. However, the fact that the space of compositional hypotheses for GNNs is so large poses computational challenges for the original modular meta-learning algorithm, which could only tackle small modular compositions and a meta-datasets of only a few hundred tasks, instead of 50.000 in our current framework.

Our contributions are the following:

1. **A model-based approach to *neural relational inference* by framing it as a *modular meta-learning* problem**. This leads to much higher data efficiency and enables the model to make inferences for which it was not originally trained.

2. **Speeding up modular meta-learning by two orders of magnitude, allowing it to scale to big datasets and modular compositions.** With respect to our previous work (Alet et al., 2018), we increase the number of modules from 6 to 20 and the number of datasets from a few hundreds to 50,000. We do so by showing we can batch computation over multiple tasks (not possible with most gradient-based meta-learning methods) and learning a proposal function that speeds up simulated annealing.

3. **We propose to leverage meta-data coming from each inner optimization during meta-training to simultaneously learn to learn and learn to optimize.** Most meta-learning algorithms only leverage loss function evaluations to propagate gradients back to a model and discard other information created by the inner loop optimization. We can leverage this "meta-data" to learn to perform these inner loop optimizations more efficiently; thus speeding up both meta-training and meta-test optimizations.

## 2 Related Work

Graph neural networks (Battaglia et al., 2018) perform computations over a graph (see recent surveys by Battaglia et al. (2018); Zhou et al. (2018); Wu et al. (2019b)), with the aim of incorporating *relational inductive biases*: assuming the existence of a set of entities and relations between them. Among their many uses, we are especially interested in their ability to model dynamical systems. GNNs have been used to model objects (Chang et al., 2016; Battaglia et al., 2016; van Steenkiste et al.,

2018; Hamrick et al., 2018), parts of objects such as particles (Mrowca et al., 2018), links (Wang et al., 2018a), or even fluids (Li et al., 2018b) and partial differential equations (Alet et al., 2019). However, most of these models assume a *fixed* graph and a single relation type that governs all interactions. We want to do without this assumption and infer the relations, as in neural relational inference (NRI) (Kipf et al., 2018). Computationally, we build on the framework of *message-passing neural networks* (MPNNs) (Gilmer et al., 2017), similar to *graph convolutional networks* (GCNs) (Kipf & Welling, 2016; Battaglia et al., 2018).

In NRI, one infers the type of every edge pair based on node states or state trajectories. This problem is related to generating graphs that follow some training distribution, as in applications such as molecule design. Some approaches generate edges independently (Simonovsky & Komodakis, 2018; Franceschi et al., 2019) or independently prune them from an over-complete graph (Selvan et al., 2018), some generate them sequentially (Johnson, 2017; Li et al., 2018a; Liu et al., 2018) and others generate graphs by first generating their junction tree (Jin et al., 2018). In our approach to NRI, we make iterative improvements to a hypothesized graph with a learned proposal function.

The literature in meta-learning (Schmidhuber, 1987; Thrun & Pratt, 2012; Lake et al., 2015) and multi-task learning (Torrey & Shavlik, 2010) is very extensive. However, it mostly involves parametric generalization; i.e., generalizing by changing parameters: either weights in a neural network, as in MAML and others variants (Finn et al., 2017; Clavera et al., 2019; Nichol et al., 2018), or in the inputs fed to the network by using LSTMs or similar methods (Ravi & Larochelle, 2017; Vinyals et al., 2016; Mishra et al., 2018; Garcia & Bruna, 2017).

In contrast, we build on our method of modular meta-learning which aims at combinatorial generalization by reusing modules in different structures. This framework is a better fit for GNNs, which also heavily exploit module reuse. Combinatorial generalization plays a key role within a growing community that aims to merge the best aspects of deep learning with structured solution spaces in order to obtain broader generalizations (Tenenbaum et al., 2011; Reed & De Freitas, 2015; Andreas et al., 2016; Fernando et al., 2017; Ellis et al., 2018; Pierrot et al., 2019). This and similar ideas in multi-task learning (Fernando et al., 2017; Meyerson & Miikkulainen, 2017), have been used to plan efficiently (Chitnis et al., 2018) or find causal structures (Bengio et al., 2019a). Notably, Chang et al. (2018) learn to tackle a single task using an input-dependent modular composition, with a neural network trained with PPO (Schulman et al., 2017), a variant of policy gradients, deciding the composition. This has similarities to our bottom-up proposal approach in section 4, except we train the proposal function via supervised learning on data from the slower simulated annealing search.

## 3 Methods

First, we describe the original approaches to neural relational inference and modular meta-learning, then we detail our strategy for meta-learning the modules for a GNN model.

### 3.1 Neural relational inference

Consider a set of $n$ known entities with states that evolve over $T$ time steps: $s_1^{1:T}, \ldots, s_n^{1:T}$. Assume that each pair of entities is related according to one of a small set of unknown relations, which govern the dynamics of the system. For instance, these entities could be charged particles that can either attract or repel each other. Our goal is to predict the evolution of the dynamical system; i.e., given $s_1^T, \ldots, s_n^T$, predict values of $s_1^{T+1:T+k}, \ldots, s_n^{T+1:T+k}$. If we knew the true relations between the entities (which pairs of particles attract or repel) it would be easy to predict the evolution of the system. However, instead of being given those relations we have to infer them from the raw observational data.

More formally, let $\mathcal{G}$ be a graph, with nodes $v_1, \ldots, v_n$ and edges $e_1, \ldots, e_{r'}$. Let $S$ be a struc-

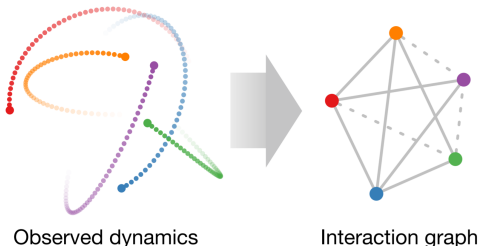

Observed dynamics      Interaction graph

Figure 2: Task setup, taken from Kipf et al. (2018): we want to predict the evolution of a dynamical system by inferring the set of relations between the entities, such as attraction and repulsion between charged particles.

ture detailing a mapping from each node to its corresponding node module and from each edge to its corresponding edge module. We can now run several steps of message passing: in each step, nodes *read* incoming messages from their neighbors and sum them, to then update their own states. The message $\mu_{ij}$ from node $i$ to $j$ is computed using the edge module determined by $S$, $m_{S_{ij}}$, which takes the states of nodes $i$ and $j$ as input, so $\mu_{ij}^t = m_{S_{ij}}\left(s_i^t, s_j^t\right)$. The state of each node is then updated using its own neural network module $m_{S_i}$ (in our experiments, this module is the same across all nodes), which takes as input the sum of its incoming messages, so

$$s_i^{t+1} = s_i^t + m_{S_i} \left( s_i^t, \sum_{j \in neigh(v_i)} \mu_{ji}^t \right) \ .$$

We apply this procedure $T$ times to get $s^{t+1}, \ldots, s^T$; the whole process is differentiable, allowing us to train the parameters of $m_{S_i}, m_{S_{ij}}$ end-to-end based on predictive loss.

In the neural relational inference (NRI) setting, the structure $S$ is latent, and must be inferred from observations of the state sequence. In particular, NRI requires *both* learning the edge and node modules, $m$, and determining which module is used in which position (finding structure $S$ for each scene). Kipf et al. (2018) propose using a variational auto-encoder with a GNN encoder and decoder, and using the Gumbel softmax representation to obtain categorical distributions. The encoder is a graph neural network that takes an embedding of the trajectory of every particle and outputs, for each node pair, a distribution over possible edge modules. The decoder samples from this factored graph distribution to get a graph representing a GNN that can be run to obtain output data. However, the probability distribution over structures is completely factored (each edge is chosen independently), which can be a poor approximation when the effects of several edges are interdependent or the graph is known *a priori* to have some structural property (such as being symmetric, a tree, or bipartite).

## 3.2 Modular meta-learning

*Meta-learning* can be seen as learning a learning algorithm. In the context of supervised learning, instead of learning a regressor $f$ with parameters $\Theta$ with the objective that $f(\boldsymbol{x}_{test}, \Theta) \approx \boldsymbol{y}_{test}$, we aim to learn an algorithm $A$ that takes a small training set $\mathcal{D}_{train} = (\boldsymbol{x}_{train}, \boldsymbol{y}_{train})$ and returns a hypothesis $h$ that performs well on the test set:

$h = A(\mathcal{D}_{train}, \Theta)$ s.t. $h(\boldsymbol{x}_{test}) \approx \boldsymbol{y}_{test}$; i.e. $A$ minimizes $\mathcal{L}(A(\mathcal{D}_{train}, \Theta)(\boldsymbol{x}_{test}), \boldsymbol{y}_{test})$ for loss $\mathcal{L}$.

Similar to conventional learning algorithms, we optimize $\Theta$, the parameters of $A$, to perform well.

Modular meta-learning learns a set of small neural network modules and forms hypotheses by composing them into different structures. In particular, let $m_1, \ldots, m_k$ be the set of modules, with parameters $\theta_1, \ldots, \theta_k$ and $\mathcal{S}$ be a set of structures that describes how modules are composed. For example, simple compositions can be adding the modules' outputs, concatenating them, or using the output of several modules to guide attention over the results of other modules.

For modular meta-learning, $\Theta = (\theta_1, \ldots, \theta_k)$ are the weights of modules $m_1, \ldots, m_k$, and the algorithm $A$ operates by searching over the set of possible structures $\mathcal{S}$ to find the one that best fits $\mathcal{D}_{train}$, and applies it to $\boldsymbol{x}_{test}$. Let $h_{S,\Theta}$ be the function that predicts the output using the modular structure $S$ and parameters $\Theta$. Then

$$A(\mathcal{D}_{train}, \Theta) = h_{S^*, \Theta} \ \text{ where } \ S^* = \arg\min_{S \in \mathcal{S}} \mathcal{L}(h_{S,\Theta}(\boldsymbol{x}_{train}), \boldsymbol{y}_{train}) \ .$$

Note that, in contrast to many meta-learning algorithms, $\Theta$ is constant when learning a new task.

At meta-training time we have to find module weights $\theta_1, \ldots, \theta_m$ that compose well. To do this, we proposed the BOUNCEGRAD algorithm (Alet et al., 2018) to optimize the modules and find the structure for each task. It works by alternating steps of simulated annealing and gradient descent. Simulated annealing (a stochastic combinatorial optimization algorithm) optimizes the structure of each task using its train split. Gradient descent steps optimize module weights with the test split, pooling gradients from each instance of a module applied to different tasks. At meta-test time, it has access to the final training data set, which it uses to perform structure search to arrive at a final hypothesis.

## 4 Modular meta-learning graph neural networks

To apply modular meta-learning to GNNs, we let $\mathbb{G}$ be the set of node modules $g_1, \ldots, g_{|\mathbb{G}|}$, where $g_i$ is a network with weights $\theta_{g_i}$, and let $\mathbb{H}$ be the set of edge modules $h_1, \ldots, h_{|\mathbb{H}|}$, where $h_i$ has weights $\theta_{h_i}$. We then apply a version of the BOUNCEGRAD method, described in the appendix. Both modular meta-learning and GNNs exhibit *combinatorial generalization*, combining small components in flexible ways to solve new problem instances, making modular meta-learning a particularly appropriate strategy for meta-learning in GNNs.

To use modular meta-learning for NRI, we create a number of edge modules that is greater or equal to the potential number of types of interactions; then with modular meta-learning we learn specialized edge modules that can span many types of behaviors with different graphs. For a new scene we infer relations by optimizing the edge modules that best fit the data and then classifying the relation according to the module used for that edge slot. This formulation of neural relational inference has a number of advantages.

First, the simulated annealing (SA) step in the BOUNCEGRAD algorithm searches the space of structures, tackling the problem directly in its combinatorial form rather than via differentiable variational approximations. Moreover, with SA, relations are inferred as a whole instead of independently; this is critical for inferring the correct relationships from short observation sequences of complex scenes, where there could be many first-order candidate explanations that roughly approximate the scene and one has to use higher-order dependencies to obtain an accurate model. For instance, if we are trying to infer the causal relationship between two variables $A$ and $B$ and we have $40\%$ probability of $A \to B$ and $60\%$ of $B \to A$, we want to express that these choices are mutually exclusive and the probability of having both edges is $0\%$ and not $24\%$.

Second, our formulation is a more direct, model-based approach. Given observational data from a new scene (task from the meta-learning perspective), we infer an underlying latent model (types of relations among the entities) by directly optimizing the ability of the inferred model to predict the observed data. This framework allows facts about the underlying model to improve inference, which improves generalization performance with small amounts of data. For example, the fact that the model class is GNNs means that the constraint of an underlying time-invariant dynamics is built into the learning algorithm. The original feed-forward inference method for NRI cannot take advantage of this important inductive bias. Another consequence of the model-based approach is that we can ask and answer other inference questions. An important example is that we can infer the existence and relational structure of unobserved entities based only on their observable effects on other entities.

However, our modular meta-learning formulation poses a substantial computational challenge. Choosing the module type for each edge in a fully connected graph requires $\binom{n}{2} = O(n^2)$ decisions; thus the search space increases as $|\mathbb{H}|^{O(n^2)}$, which too large even for small graphs. We address this problem by proposing two improvements to the BOUNCEGRAD algorithm, which together result in order-of-magnitude improvements in running time.

**Meta-learning a proposal function**   One way to improve stochastic search methods, including simulated annealing, is to improve the proposal distribution, so that many fewer proposed moves are rejected. Similar proposals have been made in the context of particle filters (Doucet et al., 2000; Mahendran et al., 2012; Andrieu & Thoms, 2008). One strategy is to improve the proposal distribution by treating it as another parameter to be meta-learned (Wang et al., 2018b); this can be effective, but only at meta-test time. We take a different approach, which is to treat the current structures in simulated annealing as training examples for a new proposal function. Note that to train this proposal function we have plenty of data coming from search at meta-training time. In particular, after we evaluate a batch of tasks we take them and their respective structures as ground truth for a batch update to the proposal function. As the algorithm learns to learn, it also learns to optimize faster since the proposal function will suggest changes that tend to be accepted more often, making meta-training (and not only meta-testing) faster, making simulated annealing structures better, which in turn improves proposal functions. This virtuous cycle is similar to the relationship between the fast policy network and the slow MCTS planner in AlphaZero (Silver et al., 2017), analogous to our proposal function and simulated annealing optimization, respectively.

Our proposal function takes a dataset $\mathcal{D}$ of state transitions and outputs a factored probability distribution over the modules for every edge. This predictor is structurally equivalent to the encoder

of Kipf et al. (2018). We use this function to generate a proposal for SA by sampling a random node, and then using the predicted distribution to resample modules for each of the incoming edges. This *blocked Gibbs sampler* is very efficient because edges going to the same node are highly correlated and it is better to propose a coherent set of changes all at once. To train the proposal function, it would be ideal to know the true structures associated with the training data. Since we do not have access to the true structures, we use the best proxy for them: the current structure in the simulated annealing search. Therefore, for each batch of datasets we do a simulated annealing step on the training data to decide whether to update the structure. Then, we use the current batch of structures as target for the current batch of datasets, providing a batch of training data for the proposal function.

Mixtures of learning-to-learn and learning-to-optimize (Li & Malik, 2016) have been made before in meta-learning in the context of meta-learning loss functions (Yu et al., 2018; Bechtle et al., 2019). Similarly, we think that other metadata generated by the inner-loop optimizations during meta-training could be useful to other few-shot learning algorithms, which could be more efficient by simultaneously learning to optimize. In doing so, we could get could get meta-learning algorithms with expressive and non-local, but also fast, inner-loop adaptations.

**Batched modular meta-learning**  From an implementation standpoint, it is important to note that, in contrast to most gradient-based meta-learning algorithms ( Zintgraf et al. (2018) being a notable exception), modular meta-learning does not need to change the weights of the neural network modules in its inner loop. This enables us to run the same network for many different datasets in a batch, exploiting the parallelization capabilities of GPUs and with constant memory cost for the network parameters. Doing so is especially convenient for GNN structures. We use a common parallelization in graph neural network training, by creating a *super-graph* composed of many graphs, one per dataset. Creating this graph only involves minor book-keeping, by renaming vertex and edges. Since both edge and node modules can run all their instances in the same graph in parallel, they will parallelize the execution of all the datasets in the batch. Moreover, since the graphs of the different datasets are disconnected, their dynamics will not affect one another. In practice, this implementation speeds up both the training and evaluation time by an order of magnitude. Similar book-keeping methods are applicable to speed up modular meta-learning for structures other than GNNs.

# 5   Experiments

We implement our solution in PyTorch (Paszke et al., 2017), using the Adam optimizer (Kingma & Ba, 2014); details and pseudo-code can be found in the appendix and code can be found at `https://github.com/FerranAlet/modular-metalearning`. We follow the choices of Kipf et al. (2018) whenever possible to make results comparable. Please see the arxiv version for complete results.

We begin by addressing two problems on which NRI was originally demonstrated, then show that our approach can be applied to the novel problem of inferring the existence of unobserved nodes.

## 5.1   Predicting physical systems

Two datasets from Kipf et al. (2018) are available online (`https://github.com/ethanfetaya/NRI/`); in each one, we observe the state of dynamical system for 50 time steps and are asked both to infer the relations between object pairs and to predict their states for the next 10 time steps.

**Springs:** a set of 5 particles move in a box with elastic collisions with the walls. Each pair of particles is connected with a spring with probability 0.5. The spring will exert forces following Hooke's law. We observe that the graph of forces is symmetric, but none of the algorithms hard-code this fact.

**Charged particles:** similar to **springs**, a set of 5 particles move in a box, but now all particles interact. A particle is set to have positive charge with probability 0.5 and negative charge otherwise. Particles of opposite charges attract and particles of the same charge repel, both following Coulomb's law. This behavior can be modeled using two edge modules, one which will pull a particle $i$ closer to $j$ and another that pushes it away. We observe that the graph of attraction is both symmetric and bipartite, but none of the algorithms hard-codes this fact.

|  | Springs | | Charged | |
|---|---|---|---|---|
| Prediction steps | 1 | 10 | 1 | 10 |
| Static | 7.93e-5 | 7.59e-3 | 5.09e-3 | 2.26e-2 |
| LSTM(single) | 2.27e-6 | 4.69e-4 | 2.71e-3 | 7.05e-3 |
| LSTM(joint) | 4.13e-8 | 2.19e-5 | 1.68e-3 | 6.45e-3 |
| NRI (full graph) | 1.66e-5 | 1.64e-3 | **1.09e-3** | 3.78e-3 |
| (Kipf et al., 2018) | **3.12e-8** | **3.29e-6** | 1.05e-3 | 3.21e-3 |
| **Modular meta-l.** | 3.13e-8 | 3.25e-6 | 1.03e-3 | **3.11e-3** |
| NRI (true graph) | 1.69e-11 | 1.32e-9 | 1.04e-3 | 3.03e-3 |

Table 1: Prediction results evaluated on datasets from Kipf et al. (2018), including their baselines for comparison. Mean-squared error in prediction after $T$ steps; lower is better. We observe that our method is able to either match or improve the performance of the auto-encoder based approach, despite it being close to optimal.

| Model | Springs | Charged |
|---|---|---|
| Correlation(data) | 52.4 | 55.8 |
| Correlation(LSTM) | 52.7 | 54.2 |
| (Kipf et al., 2018) | **99.9** | 82.1 |
| **Modular meta-l.** | **99.9** | **88.4** |
| Supervised | 99.9 | 95.0 |

Table 2: Edge type prediction accuracy. Correlation baselines try to infer the pairwise relation between two particles on a simple classifier built upon the correlation between the temporal sequence of raw states or LSTM hidden states, respectively. The supervised gold standard trains the encoder alone with the ground truth edges. Our work matches the gold standard on the springs dataset and halves the distance between the variational approach and the gold standard in the charged particles domain.

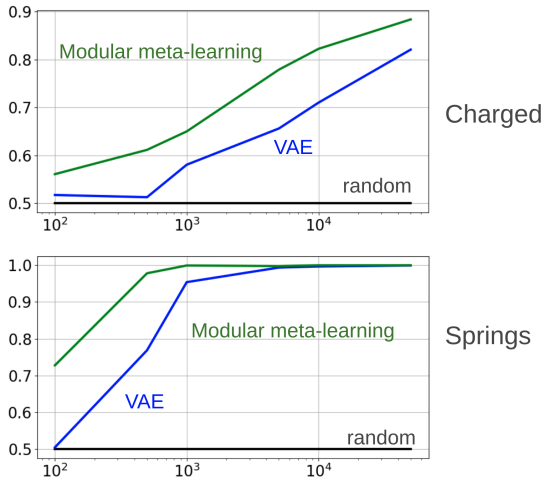

Figure 3: Accuracy as a function of the training set size (note the logarithmic axis). By being model-based, modular meta-learning is around 3-5 times more data efficient than the variational approach.

Our main goal is to recover the relations accurately just from observational data from the trajectories, despite having no labels for the relations. To do so we minimize the surrogate goal of trajectory prediction error, as our model has to discover the underlying relations in order to make good predictions. We compare to 4 baselines and the novel method used by Kipf et al. (2018). Two of these baselines resemble other popular meta-learning algorithms that do not properly exploit the modularity of the problem: feeding the data to LSTMs (either a single trajectory or the trajectory of all particles) is analogous to recurrent networks used for few-shot learning (Ravi & Larochelle, 2017) and using a graph neural network with only one edge to do predictions is similar to the work of Garcia & Bruna (2017) to classify images by creating fully connected graphs of the entire dataset. To make the comparisons as fair as possible, all the neural network architectures (with the encoder in the auto-encoder framework being our proposal function) are exactly the same.

Prediction error results (table 1) for training on the full dataset indicate that our approach performs as well as or better than all other methods on both problems. This in turn leads to better edge predictions, shown in table 2, with our method substantially more accurately predicting the edge types for the charged particle domain. By optimizing the edge-choices jointly instead of independently, our method has higher capacity, thus reaching higher accuracies for charged particles. Note that the higher capacity also comes with higher computational cost, but not higher number of parameters (since the architectures are the same). In addition, we compare generalization performance of our method and the VAE approach of Kipf et al. (2018) by plotting predictive accuracy as a function of the number of meta-training tasks in figure 3. Our more model-based strategy has a built-in inductive bias that makes it perform significantly better in the low-data regime.

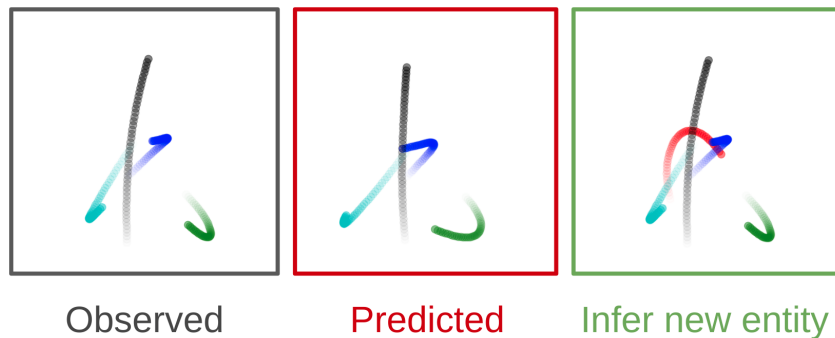

| Observed | Predicted | Infer new entity |

Figure 4: We observe the trajectories shown in the black box and notice that they differ from the predictions of our model (red box). We can then hypothesize models with an additional, unseen, entity (red particle in green box) that is pulling the cyan particle higher and the black and blue particle towards the right. Conditioning the trajectory of the particle on those predicted by our model, we make a good estimate of the position of the unseen particle.

## 5.2 Inferring unseen nodes

In many structured learning problems, we can improve the quality of our predictions by adding additional latent state. For example, in graphical models, adding "hidden cause" nodes can substantially reduce model complexity and improve generalization performance. In NRI, we may improve predictive accuracy by adding an additional latent object to the model, represented as a latent node in the GNN. A famous illustrative example is the discovery of Neptune in 1846 thanks to mathematical predictions based on its gravitational pull on Uranus, which modified its trajectory. Based on the deviations of Uranus' trajectory from its theoretical trajectory had there been no other planets, mathematicians were able to guess the existence of an unobserved planet and estimate its location.

By casting NRI as modular meta-learning, we have developed a model-based approach that allows us to infer properties beyond the edge relations. More concretely, we can add a node to the graph and optimize its trajectory as part of the inner-loop optimization of the meta-learning algorithm. We only need to add the predicted positions at every time-step $t$ for the new particle and keep the same self-supervised prediction loss. This loss will be both for the unseen object, ensuring it has a realistic trajectory, and for the observed objects, which will optimize the node state to influence the observed nodes appropriately.

In practice, optimizing the trajectory is a very non-smooth problem in $\mathbb{R}^{4 \times T}$ ($T$ is the length of the predicted trajecories) which is difficult to search. Instead of searching for an optimal trajectory, we optimize only the initial state and determine the following states by running our learned predictive model. However, since small perturbations can lead to large deviations in the long run, the optimization is highly non-linear. We thus resort to a combination of random sampling and gradient descent, where we optimize our current best guess by gradient descent, but keep sampling for radically different solutions. Detailed pseudo-code for this optimization can be found in the appendix. We illustrate this capability in the springs dataset, by first training a good model with the true edges and then finding the trajectory of one of the particles given the other four, where we are able to predict the state with an MSE of 1.09e-3, which is less than the error of some baselines that saw the entire dynamical system up to 10 timesteps prior, as seen in table 1.

## 6 Conclusion

We proposed to frame relational inference as a *modular meta-learning* problem, where neural modules are trained to be composed in different ways to solve many tasks. We demonstrated that this approach leads to improved performance with less training data. We also showed how this framing enables us to estimate the state of entities that we do not observe directly. To address the large search space of graph neural network compositions within modular meta-learning, we meta-learn a proposal function that speeds up the inner-loop simulated annealing search within the modular meta-learning algorithm, providing one or two orders of magnitude increase in the size of problems that can be addressed.

**Acknowledgments**

We gratefully acknowledge support from NSF grants 1523767 and 1723381; from AFOSR grant FA9550-17-1-0165; from ONR grant N00014-18-1-2847; from the Honda Research Institute; and from SUTD Temasek Laboratories. Any opinions, findings, and conclusions or recommendations expressed in this material are those of the authors and do not necessarily reflect the views of our sponsors.

The authors want to thank Clement Gehring for insightful discussions and his help setting up the experiments, Thomas Kipf for his quick and detailed answers about his paper and Maria Bauza for her feedback on an earlier draft of this work.

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
