[Supplementary Material]

# Appendix for:
# Neural Relational Inference
# with Fast Modular Meta-learning

**Ferran Alet, Erica Weng, Tomás Lozano Pérez, Leslie Pack Kaelbling**
MIT Computer Science and Artificial Intelligence Laboratory
{alet,ericaw,tlp,lpk}@mit.edu

## A  Experimental setup

To eliminate the amount of hyperparameters from modular meta-learning regarding the temperature schedule of Simulated Annealing we decided to auto-adjust the temperature based on the current mean-squared error loss; which essentially turns Simulated Annealing into MCMC. There are a few options how to turn the loss into the temperature, we tried 3 of them: directly using the loss as the temperature (which assume losses for different samples are perfectly correlated), using the loss divided by the number of samples (which assume they are independent). However, we went for the middle ground by computing the $R^2$ coefficient (the correlation squared) between the error in one timestep $t$ and the error on the following timestep $t + 1$, equivalent to adding a linear model on the residual as corrector and then using the loss divided by the number of samples times $(1 - R^2)$; so the temperature parameter was self-adjusted as $MSE/(F \cdot N \cdot T \cdot (1 - R^2))$ where $MSE$ is the mean-squared error loss across all datasets, $F = 4$, the number of features, $N = 5$, the number of particles, $T = 50$ the number of time-steps.

Before running all the experiments, we chose the relevant hyperparameters on runs using 10000 datasets on both springs and charged. We tried the following learning rates for Adam (Kingma & Ba, 2014): $3e - 2, 1e - 2, 3e - 3, 1e - 3, 3e - 4, 1e - 4, 3e - 5$; we chose $1e - 2$ for springs and $1e - 4$ for charged. Since the amount of overfitting was relatively small in these trial runs of 10k datasets, the number of epochs was decided mainly from computational constraints, running 1k epochs for the experiments with 50k datasets.

All other hyperparameters were the same as the code for Neural Relational Inference, to ensure they were as comparable as possible.

We run on pods containing 12GB of RAM, 5 CPUs and 1 nVIDIA RTX 2080 with PyTorch1.1 (Paszke et al., 2017) and CUDA 10.0;the biggest experiments took on the order of two days. We ran each experiment 2 times and averaged the results, except for the biggest number of datasets where we only ran them once.

## B  Pseudo-code for modular meta-learning in graph neural networks

**Notation**

- $\mathcal{G}$: graph, with node $n_1, \ldots, n_r$ and *directed* edges $e_1, \ldots, e_{r'}$.
- $f_{in}$: encoding function from input to graph initial states.
- $f_{out}$: decoding function from graph final states to output.
- $\mathbb{G}$: set of node modules $g_1, \ldots, g_{|\mathbb{G}|}$, where $g_i$ is a network with weights $\theta_{g_i}$.
- $\mathbb{H}$: set of edge modules $h_1, \ldots, h_{|\mathbb{H}|}$, where $h_i$ is a network with weights $\theta_{h_i}$.

- $S$: a structure, one module per node $m_{n_1}, \ldots, m_{n_r}$, one module per edge $m_{e_1}, \ldots, m_{e_{r'}}$. Each $m_{n_i}$ is a pointer to $\mathbb{G}$ and each $m_{e_j}$ is a pointer to $\mathbb{H}$.

- $\mathcal{T}^1, \ldots, \mathcal{T}^k$: set of regression tasks, from which we can sample $(x, y)$ pairs.

- $\mathrm{MP}^{(T)}(\mathcal{G}, S)(x_t) \to x_{t+1}$: message-passing function applied $T$ times, see Gilmer et al. (2017) for details.

- $\mathcal{L}(y_{target}, y_{pred})$: loss function; in our case $|y_{target} - y_{pred}|^2$.

- $P$ proposal function, a neural network that returns a factored probability distribution, with the probability for each module for each node and each edge.

- $\mathcal{L}_P(p, S)$: loss function for proposal function; in our case the cross-entropy loss function of probability $p$ to predict $S$.

- $L$: instantiations losses. This includes the actual loss value and infrastructure to backpropagate them.

- random_elt($\mathbb{S}, P$): pick element from set $\mathbb{S}$ according to a probability distribution

---

**Algorithm 1** `BounceGrad` with learned proposal function for Graph Neural Networks.

---
1: **procedure** INITIALIZESTRUCTURE($\mathcal{G}, \mathbb{G}, \mathbb{H}$)          ▷ Initialize with random modules
2:     **for** $n_i \in \mathcal{G}.nodes$ **do** $S.m_{n_i} \leftarrow$ random_elt($\mathbb{G}$)
3:     **for** $e_i \in \mathcal{G}.edges$ **do** $S.m_{e_i} \leftarrow$ random_elt($\mathbb{H}$)
4: **procedure** PROPOSECANDIDATESTRUCTURE($S, \mathcal{G}, \mathbb{G}, \mathbb{H}, p$)
5:     $C \leftarrow S$
6:     $idx \leftarrow random\_elt(\mathcal{G}.nodes)$
7:     **if** Bernouilli($1/2$) **then**     ▷ In our experiments we only have one node module and skip this branch
8:        $C.m_{n_{idx}} \leftarrow$ random_elt($\mathbb{G} \setminus C.m_{n_{idx}}, p_{n_{idx}}$)
9:     **else**           ▷ Resample incoming edges to one particular node
10:        **for** $e \in$ incoming($\mathcal{G}.nodes_{idx}$) **do**
11:           $C.m_e \leftarrow$ random_elt($\mathbb{H}, p_{m_e}$)
12:     **return** $P$
13: **procedure** EVALUATE($\mathcal{G}, S, \mathcal{L}, \boldsymbol{x}, \boldsymbol{y}$)
14:     $\boldsymbol{w} \leftarrow \mathrm{MP}^{(T)}(\mathcal{G}, S)(\boldsymbol{x})$           ▷ Running the GNN with modular structure S
15:     **return** $\mathcal{L}(\boldsymbol{y}, \boldsymbol{w})$
16: **procedure** BOUNCEGRAD($\mathcal{G}, \mathbb{G}, \mathbb{H}, \mathcal{T}^1, \ldots, \mathcal{T}^k$)     ▷ Modules in $\mathbb{G}, \mathbb{H}$ and proposal $P$ start untrained
17:     **for** $l \in [1, k]$ **do**
18:        $S^l \leftarrow$ InitializeStructure($\mathcal{G}, \mathbb{G}, \mathbb{H}$)
19:     **while** not done **do**
20:        $l \leftarrow random\_elt([1, k])$
21:        $(\boldsymbol{x}, \boldsymbol{y}) \leftarrow$ sample($\mathcal{T}^l$)           ▷ Train data
22:        $p \leftarrow P(\boldsymbol{x}, \boldsymbol{y})$        ▷ Proposal function predicts probabilities for every module slot
23:        $C \leftarrow$ ProposeCandidateStructure($S^l, \mathcal{G}, \mathbb{G}, \mathbb{H}, p$)
24:        $L_{S^l} \leftarrow$ Evaluate($\mathcal{G}, S^l, \mathcal{L}, \boldsymbol{x}, \boldsymbol{y}$)
25:        $L_C \leftarrow$ Evaluate($\mathcal{G}, C, \mathcal{L}, \boldsymbol{x}, \boldsymbol{y}$)
26:        $S^l \leftarrow$ SimulatedAnnealing($(S^l, L_{S^l}), (C, L_C)$)        ▷ Choose between $S^l$ and $C$
27:        $(\boldsymbol{x}', \boldsymbol{y}') \leftarrow$ sample($\mathcal{T}^l$)           ▷ Test data
28:        $L, L_P \leftarrow$ Evaluate($\mathcal{G}, S^l, \mathcal{L}, \boldsymbol{x}', \boldsymbol{y}'$), $\mathcal{L}_P(p, S)$
29:        **for** $h \in \mathbb{H}$ **do** $\theta_h \leftarrow$ GradientDescent($L, \theta_h$)
30:        **for** $g \in \mathbb{G}$ **do** $\theta_g \leftarrow$ GradientDescent($L, \theta_g$)
31:        $\theta_P \leftarrow$ GradientDescent($L_P, \theta_p$)
32:     **return** $\mathbb{G}, \mathbb{H}, P$           ▷ Return specialized modules and proposal function

---

## C  Pseudo-code for estimating the trajectory of an unseen node

Our algorithm for estimating the trajectory for the unseen node resorts to estimating its initial position and predicting the dynamical system from there. For any potential initial position for the unseen node, we can take the initial positions of all the nodes and simulate the system forward. We then compare the predictions with the observations, obtaining a loss function. Given this loss function, the initial position of the unseen node is found by maintaining a current best estimate and updating it via gradient descent (since the loss is end-to-end differentiable). Moreover, at every step we also sample a new candidate initial position and replace the current candidate with this new one in case it improves the loss. This prevents us from falling into local optima.

---

**Algorithm 2** Estimating the trajectory of an unseen node from its effects on observed nodes.

---

1: **procedure** EVALUATEUNSEENNODE($\mathcal{G}, \boldsymbol{x}_1^1, \boldsymbol{x}_{2:n}^{1:T}, \mathcal{L}, \boldsymbol{x}$)
2:     $\boldsymbol{x}^1 = [x_1^1, x_{2:n}^1]$
3:     $\boldsymbol{w} \leftarrow \text{MP}^{(T)}(\mathcal{G}, S)(\boldsymbol{x})$          ▷ Running the GNN with modular structure S from position $\boldsymbol{x}^1$
4:     **return** $\mathcal{L}(\boldsymbol{x}_{2:n}^{1:T}, \boldsymbol{w}_{2:n}^{1:T}), \boldsymbol{w}_1$     ▷ Return error in predictions for *seen* nodes and entire trajectory for $x_1$.
5: **procedure** FINDNODE($\mathcal{G}, \mathbb{G}, \mathbb{H}, S, \boldsymbol{x}_{2:n}^{1:T}$)    ▷ We assume modules are trained and we know the structure $S$.
6:     $\boldsymbol{x}_1^1 \leftarrow Uniform([0,1]^4)$          ▷ Sample an initial position and velocity for the unseen node
7:     **while** not done **do**
8:        $\boldsymbol{c}^1 \leftarrow Unif([0,1]^4)$          ▷ Sample a new candidate initial position and velocity
9:        $L_x, \boldsymbol{x}_1 \leftarrow$ EvaluateUnseenNode($\mathcal{G}, \boldsymbol{x}_1^1, \boldsymbol{x}_{2:n}^{1:T}, \mathcal{L}, S$)
10:       $L_c, \boldsymbol{c} \leftarrow$ EvaluateUnseenNode($\mathcal{G}, c, \boldsymbol{x}_{2:n}^{1:T}, \mathcal{L}, S$)
11:       **if** $L_c < L_x$ **then**          ▷ If random sample improves loss, change to new sample.
12:          $\boldsymbol{x}_1 \leftarrow \boldsymbol{c}$
13:       **else**
14:          $\boldsymbol{x}_1^1 \leftarrow$ GradientDescent($L_x, \boldsymbol{x}_1^1$) ▷ Loss is end-to-end differentiable w.r.t. initial position of $\boldsymbol{x}_1$.
15:     **return** $x_1$          ▷ Return estimate of the trajectory of the unseen node.

---