[Reviews · NeurIPS 2019]

Reviewer 1



This paper is quite unbalanced in two ways. Firstly the balance of space devoted to discussing background vs contributions is skewed too heavily towards discussing prior work, with too little focus on explaining the contributions of this work. Secondly, the coverage of the literature is heavily focused on graph networks and meta learning, but neglects to cover prior work on (non-graph based) modular networks and on learned proposal distributions. Towards the first imbalance, the section on lines 201-235 is by far the most important content in the paper, but is positioned almost as an afterthought to the extensive exposition of Alet et al. (2018). The paper would be much stronger if other sections were shortened and the descriptions in this region were substantially expanded (eg. Section 3 could be much shorter, the dataset descriptions in Section 5.1 could be moved to an appendix). In particular it would be useful to have a more in depth explanation of how training of the proposal and the simulated annealing steps are interleaved would be quite useful. This detail can be derived from the code and the pseudocode in the appendix, but as this is the main contribution of the paper it would be appropriate to include a complete description in the main body. As a result of the superficial coverage that this paper gives to its own contributions, I do not understand the contribution of the batched modular meta-learning section. My understanding of the commonly used strategy for batched evaluation of graph networks (where each example is potentially a different graph) is to do the following: 1. Merge all the graphs in the batch into a single combined graph by renumbering nodes to make a single large graph with many disjoint components. 2. Group the edges in the combined graph by type, effectively creating several batches of edges (one for each type). 3. Process each edge type batch by the appropriate edge module. 4. Aggregate the resulting messages and update each node. 5. Undo the merging from step 1. However, I suspect this is not what is described in the batched modular meta-learning section because 1) this is a standard trick and 2) line 225 states that "modular meta-learning does not need to change the weights of the neural network modules in its inner loop", and nothing about this batching strategy precludes updating the module parameters. I found the paragraph on lines 295-304 a bit light on details as well, but I think this is less critical to understanding the contributions of the paper. Towards the imbalance in literature coverage, I can point towards two very relevant bodies of work that ought to be acknowledged. The first is work on non-graph based modular networks, for example: - https://ai.stanford.edu/~ang/papers/emnlp12-SemanticCompositionalityRecursiveMatrixVectorSpaces.pdf - https://arxiv.org/abs/1511.06279 - https://arxiv.org/abs/1511.02799 - https://arxiv.org/abs/1611.01796 - https://arxiv.org/abs/1704.06611 And references therein. The second body of work is on adaptive particle filtering, where a very common approach is to parameterize (and learn) the proposal distribution. See, for example: - https://people.eecs.berkeley.edu/~jordan/sail/readings/andrieu-thoms.pdf - http://proceedings.mlr.press/v22/mahendran12/mahendran12.pdf The second paper includes references to some reviews on the topic. Another interesting, although perhaps less important, connection to make is between the method in this paper and Rao-Blackwellized particle filters (https://arxiv.org/abs/1301.3853). Rao-Blackwellization is applied in settings where you have an intractable problem that factors like p(x, y) = p(x|y)p(y) in such a way that p(x|y) can be computed efficiently. This is similar to the (training) setting in this paper where y is the graph structure and x is weights of the edge modules. Looking at the experiments in this paper I do not know how to see the effects of the statement on lines 69-70 about increased scale over prior works reflected in the numbers reported. The experiments are in general quite underwhelming as well since they only use synthetic datasets. I understand the desire to stay close to prior work for comparison, but demonstrating the method only on datasets specifically designed for the type of model under study makes the paper less compelling. ------------- After author response: I'd like to thank the authors for clarifying many of their contributions in their response, especially for pointing out that batching is only possible specifically because of the modular approach (in contrast to gradient based meta learning which cannot batch). I had not appreciated the implications of this when I wrote my original review.

Reviewer 2



This paper is generally well-written and makes an interesting contribution by combining modular meta-learning with GNNs. The key comparison is with Kipf 2018's NRI, and for 1 edge cases their performances are roughly equal, and for 10 edges and for train set size efficiency, this approach is better. This work is a good next step beyond the Alet modular meta-learning work, and (if I understand correctly) at least 1 of the BounceGrad improvements (learned proposals) would also apply to non-GNN modular meta-learning.

Reviewer 3



After rebuttal: ============================================================ Thank you for responding to all the comments! I think the rebuttal makes the contributions of the paper more clear, especially emphasizing the importance of learning the proposal function, though I think there is a line of work that also learns how to optimize the loss functions for meta-learning (e.g. [1, 2]). Nevertheless, learning a structured loss function should be insightful enough for the field. I hope in the future revision, the authors could conduct more thorough experimentations to demonstrate the effectiveness of the method, e.g. on a more realistic dataset, and restructure the paper in a more organized and clear way. [1] Yu, T., Finn, C., Xie, A., Dasari, S., Zhang, T., Abbeel, P., & Levine, S. (2018). One-shot imitation from observing humans via domain-adaptive meta-learning. arXiv preprint arXiv:1802.01557. [2] Chebotar, Y., Molchanov, A., Bechtle, S., Righetti, L., Meier, F., & Sukhatme, G. (2019). Meta-Learning via Learned Loss. arXiv preprint arXiv:1906.05374. Before rebuttal: ============================================================ This paper builds neural relational inference upon the modular meta-learning framework, which can capture the dependence among edges instead of modeling different edges independently. Moreover, the method also uses meta-learning to meta-learn a proposal function with self-supervision, which greatly improves the efficiency of the algorithm. While the paper is novel and tackles the problem that the previous variational neural relation inference doesn't address, the paper is poorly written and hard to understand. The authors fail to provide clear justification of their method in Sec.4 despite an algorithm summary provided in the appendix, which is also a bit hard to grasp without detailed explanation. Moreover, the experiments are also not entirely convincing. While the edge type prediction accuracy demonstrates that the proposed method is able to classify edges much more accurately than the previous NRI method in the charged particle setting, the authors should provide more experimental results to demonstrate the effectiveness of the method. Section 5.2 is also written in a confusing way with pure text. There is Figure 4 but not referenced anywhere in the paper. The authors should provide more detailed explanations on how they add the new node and how their method compares to the baselines in the unseen node setting. Overall, this paper shows an interesting combination of NRI and modular meta-learning that leads to effective joint edge inference but lacks clarity and enough experimental support.

[Author Response · NeurIPS 2019]

As pointed out by reviewers, we need to devote more time to highlight our contributions. We add here some aspects that we did not highlight with enough detail:

1. Learning a proposal function is an important contribution to meta-learning.

   (a) Two of the main applications of meta-learning are learning new tasks from few examples and learning to be fast at optimizing new functions. To the best of our knowledge, typical few-shot learning algorithms do not simultaneously learn to optimize the loss function of each task quickly. In contrast, we learn from few examples by doing structure search and speed it up by orders of magnitude by learning a proposal function.

   (b) Usually, few-shot learning algorithms either rely on local search to perform fast optimization or perform more complex, but slow, optimizations; limiting either capacity or speed. By also learning how to optimize, we can have high capacity and fast search.

   (c) During meta-training, the optimization of each task generates a lot of data (such as gradients, intermediate weights or, in our case, structures). However, most meta-learning algorithms only use the final loss performance. Learning our proposal function shows how we can profit from aggregating these other types of information.

2. Batched modular meta-learning enables shared computation between tasks.

   (a) Main idea: modular meta-learning runs the same module weights across all tasks. Therefore, if a module is present in multiple tasks, we can batch all its evaluations into a single one. This gives similar compute gains as GPUs evaluating regular neural networks in batches of examples, but now for batches of tasks as well and leads to a factor of 5 speed-up.

   (b) Coding this parallelization is very easy for many modular meta-learning compositions. For instance, as reviewer 1 points out, parallelizing GNNs is a simple and standard idea. We see this simplicity as a positive aspect.

   (c) Note that gradient-based few-shot learning algorithms cannot do this parallelization, because they change their network weights differently in each task.

3. A model-based approach to Neural Relational Inference.

   (a) We can tackle new types of problems for which we did not explicitly train. We highlight this by finding the trajectory of an unseen particle using only its influence on seen particles, which the original NRI approach can not do.

   (b) We can predict all edges jointly instead of independently of one another. This is important because many graphs have some high-level structure, such as being connected, symmetric or bipartite. Predicting them jointly adds more capacity to the search and makes it easier to unearth these high-level relations.

We will make sure to better balance our related work section to make higher emphasis on non-graph modular networks. We also agree that learning proposal functions has been studied in other fields, such as in particle filters; we thank reviewer 1 for the relevant citations, which we will certainly include.

Given the reviews, we acknowledge the paper would benefit from more clarity explaining the methods and the experiment (like section 5.2 and figure 4). We will expand both sections, if needed moving details (such as section 5.1) to the appendix. With respect to the experiments, we would like to mention the following things:

1. We do not include comparisons with the original modular meta-learning because we expect results to be similar, but training to take about one month. In particular, we believe one of the contributions of this work is making modular meta-learning practical, going from the toy datasets of 150 tasks of the original paper by Alet et al., to 50.000 in this work.

2. None of the baselines, including the original NRI method, can directly tackle the new problem of finding an unseen node via its influence on others. Thanks to our model-based approach we can make the trajectory of the unseen node part of the inner-loop optimization. To show that our results are good, we show that our prediction errors are as good or better than some of the baselines that do know the position of all nodes, showing good generalization beyond the single problem of finding relations.

[Meta-Review · NeurIPS 2019]

This paper is quite borderline. The reviewers were all learning towards acceptance, but none were willing to fight for the paper. The main concern for the paper is that the empirical performance gain over Kipf et al is quite small, and in some cases, non-existent. In the balance, I would put this paper slightly over the bar for acceptance. However, we strongly encourage the authors to try to better highlight the benefits over Kipf et al regarding unseen nodes, in the final version. We also encourage the authors to quantitatively evaluate the speed difference from Alet et al, e.g. for one iteration of the algorithm, and to address the points raised by the reviewers.